# Nephroprotective Mechanisms of SGLT2i: Beyond the Glucose-Lowering Effect

**DOI:** 10.3390/biomedicines13092123

**Published:** 2025-08-30

**Authors:** Alessio Mazzieri, Livia Maria Rita Marcon

**Affiliations:** 1Diabetes Clinic, Hospital of Città di Castello, USL Umbria 1, 06012 Perugia, Italy; 2Department of Endocrinology, Metabolic Diseases and Nutrition, ASST-Brianza, 20900 Vimercate, Italy; livia.marcon@gmail.com

**Keywords:** chronic kidney disease, diabetes mellitus, sodium–glucose cotransporter 2 inhibitors, drug repurposing

## Abstract

Chronic kidney disease (CKD) is a fast-growing cause of death worldwide. Systemic hypertension and diabetes mellitus are the major causes of kidney damage leading to a reduction in glomerular filtration rate and to urinary protein loss. Sodium–glucose cotransporter 2 inhibitors (SGLT2is) are drugs able to address both of these deleterious effects, preventing kidney damage from progressing. Initially born as hypoglycemic agents, SGLT2is subsequently proved to have not only positive metabolic but also pleiotropic effects on the kidney and the cardiovascular system. Indeed, they improve the metabolic profile, reducing uric acid, blood sugar levels, and body weight. Moreover, they exert an anti-inflammatory and antifibrotic effect, reducing endothelial dysfunction and reactive oxygen species (ROS) production. Finally, they reduce renal hyperfiltration and control blood pressure, inducing osmotic diuresis and restoring tubulo-glomerular feedback. All these metabolic, anti-inflammatory, and hemodynamic effects contribute to significantly reducing the risk of cardiorenal events, as widely demonstrated in randomized clinical trials. The pleiotropic actions of SGLT2is together with their good tolerability make them a pillar treatment of CKD regardless of the presence of diabetes mellitus. Further studies will be needed in order to expand the indications to populations previously excluded from clinical trials such as transplant recipients or glomerulonephritis patients. This narrative review aims to summarize current knowledge regarding the nephroprotective mechanisms of SGLT2is which, after initial use as a hypoglycemic agent, have assumed a pivotal role in the actual and future management of patients with CKD.

## 1. Introduction

Sodium–glucose cotransporter 2 inhibitors (SGLT2is) are a class of insulin-independent hypoglycemic agents that inhibit the reabsorption of sodium and glucose in the proximal convoluted tubules of the kidney. Their action provides increased urinary glucose excretion and mild osmotic diuresis [1], which in turn are responsible for plenty of clinical benefits. SGLT2is induce reductions in blood glucose levels while avoiding hypoglycemia, modest body weight loss, reduced blood pressure values, and lower risk of major adverse cardiovascular events (MACE). SGLT2is have already demonstrated the ability to counteract many of the pathophysiological abnormalities leading to chronic kidney disease (CKD) and heart failure (HF) onset and progression. The global prevalence of CKD and HF are, respectively, >10% and 1–3%, increasing in diabetic people to approximately 30% and 20% [2,3,4]. After decades of no significant progress in the prevention and delay of these diseases, such evidence has set SGLT2is as a pillar therapy in the treatment of both diabetic and non-diabetic patients with cardiorenal compromise due to HF or CKD [5,6,7,8].

Despite the mechanisms of action of these agents having been widely investigated, many others remain unknown and still need further investigation [9,10]. A large number of these compounds have received marketing approval, including four that are currently authorized by both the EMA and the FDA: canagliflozin, dapagliflozin, empagliflozin, and ertugliflozin.

A deeper understanding of the molecular basis leading to CKD has contributed to the development of precision medicine addressing specific targets along the pathogenetic pathway. Among the therapeutic options currently available, SGLT2is stand out since, as compounds, they are able to implement their action on most of them. In particular, the nephroprotection mechanisms offered by SGLT2is can be divided into metabolic effects; antioxidant, anti-inflammatory, and antifibrotic effects; and hemodynamic effects. These mechanisms are shown in Figure 1 and explained in detail in Table 1.

**Table 1 biomedicines-13-02123-t001:** The table shows the nephroprotective mechanisms offered by SGLT2is, divided into metabolic effects; antioxidant, anti-inflammatory, and antifibrotic effects; and hemodynamic effects. FFA, free fatty acids; HIF-1α, hypoxia-inducible factor-1 subunit alpha; HIF-2α, hypoxia-inducible factor-2 subunit alpha; RAS, renin–angiotensin system; SNS, sympathetic nervous system.

**Metabolic** **effects**	Provide a negative caloric balance, promoting blood glucose reduction and weight loss [11] Induce a metabolic shift towards FFA utilization and ketogenesis [12], promoting autophagy [13] and improvement of dyslipidemia [14] Promote a reduction in uric acid levels [15]
**Antioxidant,** **anti-inflammatory, and antifibrotic effects**	Decrease oxidative stress, increasing bioavailability of nitric oxide [16] Decrease pro-inflammatory cytokines levels [17] Reduce renal hypoxia, suppressing HIF-1α and activating HIF-2α [18] Decrease the activation of RAS [19]
**Hemodynamic effects**	Induce osmotic diuresis and transient natriuresis [20] Restore tubologlomerular feedback, reducing intraglomerular pressure and albuminuria [21] Control blood pressure by regulation of SNS [22]

The following sections of this narrative review aim to summarize the nephroprotective mechanisms of SGLT2is, which born as glucose-lowering drugs, became a pillar treatment in the management of patients with CKD.

## 2. Metabolic Effects

### 2.1. Reduction in Body Weight

SGLT2is accomplish a reduction in body weight mainly due to the glycosuric effect. In fact, a daily urinary loss of about 70–90 g of glucose further generates a daily energy loss of about 200–250 kcal per day [11]. Fluid loss may also contribute to some extent, as studies measuring body composition have shown that most of the change in total body mass consists of the loss of both water and fat mass, with minimal impact on total lean body mass [23,24]. Bioimpedance analysis confirms that, in patients treated with SGLT2is, body mass reduction in the medium term is due to a reduction in adipose tissue and to only a transient loss of extracellular fluid (−0.4 L of extracellular water per 1.73 m^2^ of body surface area at 1 month), which normalizes within 6 months, while lean tissue mass is preserved [25]. A meta-analysis of clinical trials on SGLT2i indicates an average body mass reduction of 2 kg compared to the placebo [26]. The reduction in body mass is observed especially in the first four weeks of treatment and subsequently stabilizes probably due to the activation of compensatory mechanisms such as appetite and increased endogenous glucose synthesis [27]. In fact, dapagliflozin-induced glucose and weight loss are partially counteracted by a concomitant increase in glucagon and hence glucose synthesis of 32% and 17%, respectively, which might be one of the probable mechanisms mitigating the calorie deficit and attenuating weight loss [28]. In addition to glycosuria, other possible mechanisms of action of SGLT2is on adiposity, still needing verification, are the prevention of mitochondrial dysfunction and the modulation of adiponectin levels [29]. Experimental evidence demonstrated that empagliflozin reduces inflammation in adipose tissue, increases energy expenditure and heat production, and promotes the expression of decoupled protein 1 in brown adipose tissue [30]. Another study showed that empagliflozin potentiates fatty acid beta oxidation through upregulation of MAPK pathways, reduces hepatic steatosis, and increases adiponectin levels in ApoE-/- mice [31]. Finally, several studies indicated that SGLT2i-induced weight loss is associated with reduced sympathetic nervous system (SNS) activity [32], which may be closely associated with SGLT2i-mediated weight reductions [33]. Compared to the results obtained by other drugs, the body weight reduction induced by SGLT2i remains modest in amount, though contributing clinically to a significant reduction in visceral and subcutaneous adipose tissue and to a decrease in lipotoxicity, with consequent prevention of impaired renal function [34].

### 2.2. Effects on Dyslipidemia: Reduction in LDLc and Increase in HDLc

The glycosuria induced by SGLT2i leads to a modification of the substrate mostly used to obtain cellular energy, shifting from a metabolism mainly based on glucose to free fatty acid (FFA) oxidation [35]. The switch contributes to a drop in toxic lipid metabolite levels inside the cells, including podocytes, mesangial cells, and proximal tubular cells, with a final nephroprotective effect [36]. In addition, the reduction in subcutaneous and visceral adipose tissue induced by SGLT2i treatment improves hepatic steatosis and insulin sensitivity while decreasing plasma levels of triglycerides and total cholesterol, with beneficial qualitative changes on LDL (e.g., less atherogenic lipid profile; reduction in triglycerides and small, dense LDL particles [37]) and increases in HDL particles [14].

### 2.3. Metabolic Reprogramming: Ketogenesis

Through the process of β-oxidation in the liver, FFAs are converted to acetyl-CoA, which in turn are further transformed into ketone bodies, resulting in a metabolic condition similar to a prolonged fasting [38]. Moreover, lipid oxidation is associated with the reversal of lipotoxicity and indirectly with the improvement of muscle insulin sensitivity and β-cell function [39]. Inhibition of the SGLT2 cotransporter and the subsequent glycosuria enhance FFA oxidation and significantly raise plasma concentrations of ketone bodies compared to baseline conditions [12]. Indeed, available evidence suggests that plasma levels of ketone bodies increase during prolonged treatment with SGLT2is, and although mean levels of ketone bodies are only modestly elevated, they may increase in the millimolar range in many patients (up to 5 mmol/L vs. 100–600 µmol/L in normal physiological conditions [40,41]), particularly those with insulinopenia [12]. Different mechanisms induced by the action of SGLT2is and ketogenesis therefore contribute to nephroprotection. Ketones, being a more energy-efficient substrate than glucose, help reduce oxygen consumption and, at the same time, inhibit rapamycin complex 1, which is associated with the development of kidney damage [42]. Moreover, the stabilization of the energy-consuming processes in the tubular epithelial cells reduce adenosine triphosphate (ATP) consumption and direct the energy towards cellular repair and autophagy mechanisms [42]. At the same time, the glycosuric effect creates a “nutrient deprivation” signaling that upregulates adenosine monophosphate-activated protein kinase (AMPK) [13], a key regulator of cellular autophagy. Lastly, treatment with SGLT2is reduces the loss of podocytes, improves podocyte function, and reduces albuminuria as the switch from glucose to FFA oxidation could decrease the lipid content in these cells. SGLT2is, inducing carbohydrate and energy deficit, produce a metabolic transition from carbohydrate to lipid use and stimulate ketogenesis, with consequent activation of cellular autophagy processes and reduction in oxidative stress and proinflammatory and fibrotic processes in the kidneys and in the whole body. However, it is evident that ketogenesis is a complex process and its health-related outcomes must be balanced with negative effects. Apart from the possible ketoacidosis events, particularly in the case of insulinopenia, an excess of plasma ketones can lead to senescence cell and organ damage perhaps due to an increase in lipids or lipoproteins [43].

### 2.4. Reduction in Plasma Uric Acid Levels

Treatment with SGLT2is contributes to the reduction in uricemia through increased urinary elimination of uric acid. However, these mechanisms are not yet fully identified. The increase in glycosuria induced by SGLT2 inhibition reduces urate uptake into the proximal convoluted tubule (PCT) via glucose transporter 9b (GLUT9b) [15]. Indeed, the uricosuria induced by SGLT2 inhibition is attributed to increased concentration of luminal glucose in the PCT that competes with urate for GLUT9b [44]. In a meta-analysis of 62 clinical trials involving SGLT2i, Zhao et al. showed that all SGLT2is significantly decreased uric acid levels compared with the control (reduction by 35–45 μmol/L). This effect was observed within a few days of study initiation and up to 2 years [45]. Although the goal of a clinical benefit from reducing uric acid levels is not yet fully clear, much evidence reports a strong association between uric acid concentrations and cardiovascular outcomes [46]. An exploratory analysis of the EMPA-REG OUTCOME [47] suggested that variations in uric acid levels contributed to a minor extent to a reduction in cardiovascular death rate.

## 3. Antioxidant, Anti-Inflammatory, and Antifibrotic Effects

At the level of the renal parenchyma, SGLT2is can modify systemic inflammation through various effects: the reduction in oxidative stress mediated by hyperglycemia and reduced nitric oxide production, the reduction in body weight, and the activation of the renin–angiotensin–aldosterone system (RAAS) with the consequent hemodynamic influences [30,48,49]. Both in kidneys of diabetic rats after 4 weeks of empagliflozin therapy [50] and in cultured human proximal tubular cells after treatment with dapagliflozin [51], reduced generation of advanced glycation end-products (AGEs) and suppression of the AGE-AGE receptor axis has been demonstrated. Treatment with SGLT2is has been shown to increase the bioavailability of nitric oxide through the upregulation of genes involved in its synthesis, alleviating endothelial dysfunction, endothelial inflammation, and reactive oxygen species (ROS) production [16]. In addition, SGLT2is have been demonstrated to inhibit the expression of monocyte chemoattractant protein-1 (MCP-1), an important factor leading to inflammation, fibrosis, and tubular atrophy [52]. Several studies have shown that renal fibrosis is reduced by SGLT2i primarily through the reduction in hypoxia, inflammation, and oxidative stress and the simultaneous activation of RAAS, all of which are associated with the renal fibrosis process [19,53]. SGLT2is are able to suppress the transforming growth factor beta 1 (TGFB1)-related inflammatory cascade in damaged proximal tubular cells and to reduce the expression of hypoxia-inducible factor-1 subunit alpha (HIF-1α) and nuclear factor kappa-light-chain-enhancer of activated B cells (NF-kB), underlining how the antifibrotic effects of SGLT2is could be in direct connection with the reduction in renal hypoxia [53]. In this regard, SGLT2is also activate hypoxia-inducible factor-2 subunit alpha (HIF-2α), enhancing autophagy and mitophagy with consequent removal and renewal of damaged mitochondria and peroxisomes, which are major sources of oxidative stress [18]. In addition, SGLT2is have been shown to reduce circulating levels of interleukin 6 (IL-6), tumor necrosis factor 1 (TNF-1) receptor, and matrix metalloproteinase-7 (MMP-7) and fibronectin-1, key molecules in the formation process of diabetic kidney disease [17]. Finally, SGLT2is reduce kidney fibrosis, decreasing the pro-fibrotic molecules involved in the renin–angiotensin system (RAS), which is activated by default in CKD [19].

## 4. Hemodynamic Effects

### 4.1. Osmotic Diuresis and Natriuresis

The mechanism of action of SGLT2is includes increased urinary glucose excretion leading to osmotic diuresis, with a transient increase in natriuresis, also caused by reduced sodium reabsorption in the proximal tubule, demonstrated in both animal and human studies. SGLT2i-induced glycosuria and natriuresis contribute to the improvement of the hemodynamic functioning of periglomerular and glomerular vessels, the prevention of salt-sensitive hypertension, and the reduction in volume overload [24]. Because glucose and sodium reuptake in the proximal convoluted tubule is coupled, SGLT2 inhibition is associated with a mildly negative sodium–water balance and an initial decrease in extracellular fluid and plasma volume [20]. Furthermore, in relation to the improvement of the water balance and the promoted natriuresis, it was found that SGLT2 inhibition can indirectly lead to a decrease in the enzymatic activity of sodium–hydrogen exchanger 1 (NHE1) and 3 (NHE-3). In particular, SGLT2is can exert a direct action on NHE3 transporters by promoting the phosphorylation of two specific serine sites at the enzyme level, promoting urinary sodium loss [54]. SGLT2i-induced reduction in cytoplasmic sodium concentration increases mitochondrial calcium levels, further contributing to nephroprotective mechanisms [54]. The acute natriuretic effect of SGLT2 inhibition typically manifests with an increase in urinary volume of 300 mL per day for the first 2–3 days. It takes several weeks to return to pre-treatment urinary volumes with the restoration of a new sodium–water balance and a reduction of approximately 7% in plasma volume (with a wide range between individuals ranging from 5 to 12%) within 3 months of treatment [20]. The natriuretic effect is also related to a reduction in cutaneous sodium content. A reduction in skin sodium content after 6 weeks of treatment with dapagliflozin was demonstrated in a study in patients with T2DM [55]. The significance of this change is not yet known, but the increase in skin sodium content could act as a sodium reservoir in case of water overload and has been associated with hypertension and left ventricular hypertrophy in patients with chronic kidney disease [56]. It has also been hypothesized that osmotic diuresis induced by SGLT2i therapy results in greater clearance of free water from interstitial space compared to systemic circulation [57]. In fact, SGLT2i-induced fluid loss predominantly affects the interstitial tissue, resulting in reduced renal congestion and interstitial edema, without a significant impact on intravascular volume and less activation of the sympathetic system [58]. Considering the data available regarding SGLT2i use, volume depletion resulting from glycosuria and natriuresis is more common: in elderly subjects, in subjects treated with glycosuric therapy at maximum dose, in subjects on loop diuretics, and in those with reduced renal function [59]. However, volume-related adverse events, such as hypotension, were no more common in the SGLT2i group than in the placebo group in the EMPA-REG OUTCOME study [5] or the DAPA-HF study [60]. It is also likely that natriuresis and a reduction in plasma volume are protective against the development of heart failure and could explain, at least in part, a reduction in the risk of hospitalization for heart failure observed in CVOTs, as in the CREDENCE study [61] and in the DAPA-HF study [60]. Glycosuria and natriuresis and the consequent negative water balance of about 700 mL achieved within the first 48–72 h after the first dose, are the effects probably responsible for the rapid changes in blood pressure induced by SGLT2i, with an average reduction of approximately 3–7 mmHg in systolic blood pressure (SBP) and of 1–3 mmHg in diastolic blood pressure (DBP) [62]. The reduction in blood pressure is generally observed regardless of the presence of hypertension and even in patients with lower estimated glomerular filtration rate (eGFR) values and is followed by a slowing of the progression of kidney disease. Given the modest impact of glycosuria and natriuresis on blood pressure, other protective factors, described later, seem to play a more impactful role in the blood pressure reduction [63].

### 4.2. Modulation of Tubuloglomerular Feedback

Under physiological conditions, SGLT2 is responsible for ≤5% of total renal sodium reabsorption. This percentage increases up to 15% in conditions of hyperglycemia due to the upregulation of SGLT2 and SGLT1 in the kidney [64]. This change leads to a drop in sodium influx to the dense macula. The decrease in sodium content is perceived by the juxtaglomerular apparatus as a reduction in the effective circulating plasma volume, inducing vasodilation of the afferent arteriole, an increase in intraglomerular pressure, and hyperfiltration [65]. Hyperfiltration is thought to be a risk factor for the progression of renal disease acting directly on the single nephron [66]. SGLT2i increases distal renal sodium influx, thereby increasing glomerular afferent arteriole tone and reducing hyperfiltration [67]. In patients with T2DM, SGLT2i therapy induces an acute dose-dependent reduction in eGFR of approximately 5 mL/min/1.73 m^2^ in the initial weeks and slowly returning towards pretreatment values over the next 3–9 months [68]. It is likely that the mechanism of this acute reduction in eGFR is due to the fact that SGLT2is, through increased urinary sodium excretion reaching the dense macula, restore tubuloglomerular feedback and contribute to vasoconstriction of the afferent arteriole with the reduction in intraglomerular pressure [69,70]. Therefore, the reduction in intraglomerular pressure encompasses the main nephroprotective effects of SGLT2i, leading both to an improvement in albuminuria and to a slower rate of decline in renal function [21]. With respect to the improvement of albuminuria, a meta-analysis of 48 randomized controlled trials (RCTs) of SGLT2i used for ≥12 weeks and including >58,000 patients reported a reduction in the urinary albumin/creatinine ratio (weighted mean difference −14.6 mg/g,), with a greater effect in subjects with a higher ratio at baseline, a higher eGFR at baseline, and a longer duration of treatment. Specifically, the risk of developing microalbuminuria was reduced by 31%, the risk of macroalbuminuria by 51%, the risk of worsening nephropathy by 27%, and the risk of end-stage renal disease (ESRD) by 30% [71]. In large cardiovascular outcome studies, the introduction of SGLT2i in patients receiving RAAS blocker therapy reduced albuminuria and slowed the progression from microalbuminuria to macroalbuminuria [71]. In this regard, it should be specified that the mechanism of action of SGLT2is is added to that of RAAS blockers, which on the other hand, contribute to nephroprotection, reducing intraglomerular pressure by vasodilation of the efferent arteriole, leading to reductions in both intraglomerular hypertension and renal hyperfiltration [66]. In essence, while RAAS blockers primarily focus on reducing efferent arteriolar constriction, SGLT2 inhibitors offer a broader approach by constricting afferent arterioles and reducing efferent arteriolar resistance, leading to a reduction in intraglomerular pressure and potentially a greater degree of kidney protection [72]. It is interesting to note that the level of long-term nephroprotection provided by RAAS inhibition is related to the extent of the initial reduction in eGFR (the higher the initial eGFR reduction, the better the long-term nephroprotection) [73]. It is not yet clear whether the initial SGLT2i-induced decline in eGFR can be interpreted in the same way. However, concomitant use of RAAS inhibitors or loop diuretics with SGLT2i appears to be associated with a higher risk of acute kidney injury (AKI) [73]. From secondary analyses of the available trials, the risk of AKI under SGLT2i use appears to be minimal and comparable to that obtained with placebo, though data may be affected by a possible bias in the selection of the patients enrolled. Some authors, based on both randomized and observational studies, even argue that SGLT2is reduce rather than increase the risk of AKI [74].

### 4.3. Neuroregulation of Blood Pressure

SNS hyperactivity is known to be one of the potential mechanisms involved in the pathophysiology of hypertension [75,76]. Experimental in vitro and in vivo studies provide evidence of a cross-talk between SNS and SGLT2 regulation [77]. Sympathetic nerve endings innervate the proximal tubules of the kidney, where they have been shown to regulate the expression of not only transporters such as sodium–hydrogen exchanger 3 but also of SGLT2 [78]. Moreover, due to the presence of SGLT2 receptors in the rostral ventrolateral medulla, SGLT2i may alleviate hypertension by weakening the activity of the sympathetic nervous system at the central level [22]. Treatment with the most experienced SGLT2i (canagliflozin, dapagliflozin, and empagliflozin) has in fact demonstrated that attenuation of sympathetic tone is one of the mechanisms of blood pressure reduction induced by glycosuric therapy [79]. Additionally, the recent literature indicates that SGLT2 receptors are present in the nucleus tractus solitarius (nTS), a central station for the body’s cardiorespiratory afferent inputs. The not fully clear action of SGLT2is on nTS might be another mechanism by which this compound could regulate blood pressure [22].

## 5. Safety Profile of SGLT2i

Regarding pharmacological safety aspects, the most frequently associated adverse events associated with the use of SGLT2is are urogenital infections, with a documented risk two to six times greater than that of the placebo, depending on the study. This adverse effect had a mean prevalence in clinical trials of approximately 10% in women and 5% in men [5] and was associated with treatment discontinuation or an adverse effect on quality of life. The urogenital infections in question are related to the iatrogenic process of glycosuria and are generally fungal (particularly with Candida species) [80]. However, most of these cases are mild and respond to standard treatment with single-dose topical or oral antifungals [81]. With regard to other adverse effects, a meta-analysis that included more than 78,000 participants, diabetic and non-diabetic, with a previous cardiovascular history or at high cardiovascular risk, demonstrated the superiority of SGLT2i over a placebo in reducing the risk of acute renal failure (particularly when empagliflozin and dapagliflozin were used). In the same meta-analysis, SGLT2is increased the risk of euglycemic ketoacidosis at the same time. This is a rare adverse effect, found in 1 in 1000 SGLT2i users, related to volume depletion situations that can be avoided by stopping their administration in the case of prolonged fasting, surgery, or critical illness. Contrary to initial reports, treatment with SGLT2i has shown no significant effect on the incidence of amputations, bone fractures, thromboembolic events, or hypoglycemia [82,83]. In addition, in view of the site of renal action and associated osmotic diuresis, there is general concern about the potential effects of SGLT2i on electrolyte equilibria and volume depletion [84]. However, the available evidence suggests that SGLT2i is associated with only slight changes in serum electrolyte levels [85,86,87]. Although a negative sodium balance can be postulated during SGLT2i administration, no significant changes in serum sodium levels have been reported in clinical trials [88]. However, although mild sodium and water depletion contributes to the hemodynamic effects of SGLT2i, the evidence currently available on SGLT2i has also shown only modest variations in serum potassium and magnesium concentrations [85,86,87]. Cases of mild hyperkalemia have been associated with high doses of canagliflozin [89] but have not been seen with empagliflozin [90] and dapagliflozin [91]. It is unclear whether this effect on potassium is caused by small transient changes in GFR or volume status or by an improvement in insulin resistance [92]. However, even if the increase in serum potassium concentration is modest, close monitoring is indicated in patients with renal impairment or in patients undergoing therapy predisposed to hyperkalemia, such as RAAS blockers [93]. SGLT2 inhibitors can also increase serum magnesium levels in diabetic patients [87] through a number of mechanisms that are still not fully understood, including a reduction in insulin resistance [94]. Considering that hypomagnesemia in diabetic patients is associated with an increased cardiovascular risk [95,96], the increase in serum magnesium induced by SGLT2i could be among the cardioprotective effects associated with this pharmacological class.

## 6. Drug Repurposing of SGLT2i for Nephroprotection

The concept of drug repurposing [97] refers to the strategy used to discover new applications for already approved, abandoned, or experimental drugs that are outside the scope of their original indication. The strategy has several advantages over the development of completely new drugs. In particular, the time to development and the risk of failure are shorter because the existing compound will have been shown to be safe in preclinical models and in humans. The fundamental premise of drug repurposing is to overcome the main mechanism of action of a drug and exploit its pleiotropic effects in search of a clinical benefit [97]. This concept, which can be found in the pharmacological evolution of various therapies, is also clearly applied to the SGLT2i class in the field of nephroprotection. Although they were originally designed as hypoglycemic drugs promoting urinary glucose excretion, it was later appreciated that SGLT2i were, in fact, a valid therapy for reducing cardiorenal risk and slowing the progression of CKD, regardless of diabetic disease [98]. In particular, the nephroprotective effects of SGLT2i have been demonstrated and confirmed by a multitude of clinical trials, starting with cardiovascular outcome trials (CVOTs) [99]. The study “Canagliflozin and Renal Events in Diabetes with Established Nephropathy Clinical Evaluation” (CREDENCE) [61] was the first study conducted with an SGLT2i with a renal primary endpoint (ESRD, doubling of creatinine level, or death from renal or cardiovascular causes). The study enrolled 4401 patients with type 2 diabetes mellitus and albuminuric CKD (eGFR, CKD-EPI, between 30 and 90 mL/min/1.73 m^2^ and AUCR between 300 and 5000 mg/g), on RAAS blocker therapy. Canagliflozin achieved a 30% reduction in the relative risk of meeting the primary endpoint and a 32% reduction in the relative risk of progression of ESRD. Therefore, in 2020, canagliflozin was approved by the FDA to reduce the risk of end-stage renal disease, doubling of serum creatinine levels, cardiovascular death, and hospitalization in adults with type 2 diabetes, and diabetic kidney disease with albuminuria. Following the CREDENCE study, the study “Dapagliflozin and Prevention of Adverse Outcomes in Chronic Kidney Disease (DAPA-CKD)” [8] involved a non-diabetic CKD population in addition to diabetic patients (67.5% of study participants). The study enrolled 4304 patients with DKD or CKD (mean eGFR 43.1 ± 12.4 mL/min and albuminuria/creatininuria ratio between 200 and 5000 mg/g). The results showed that dapagliflozin achieved a 39% reduction in the relative risk of meeting the primary endpoint and a 44% reduction in the relative risk of sustained 50% reduction in eGFR, ESRD, or death from renal or cardiovascular causes [8]. Based on these findings, in 2021, dapagliflozin was approved by the FDA to reduce the risk of prolonged eGFR decline, end-stage renal disease, cardiovascular death, and hospitalization for heart failure in adults with CKD at risk of progression, regardless of type 2 diabetes status. More recently, “The Study of Heart and Kidney Protection With Empagliflozin” (EMPA-KIDNEY) [100] was designed to evaluate the effect of empagliflozin on the progression of renal and cardiovascular disease in a wide variety of patients with chronic kidney disease, with and without diabetes (54% of patients were non-diabetic), with and without albuminuria (20% were normoalbuminuric). The study enrolled 6609 patients with DKD or CKD (eGFR 20–45 mL/min/1.73 m^2^ or eGFR 45–90 mL/min/1.73 m^2^ and UACR ≥ 200 mg/g) and showed that empagliflozin reduced the relative risk of ESRD progression or cardiovascular death by 28% compared to the placebo. Similar to dapagliflozin, in 2023, empagliflozin received FDA approval to reduce the risk of prolonged eGFR decline, end-stage renal disease, cardiovascular death, and hospitalization in adults with CKD at risk of progression, regardless of type 2 diabetes status [101]. A secondary analysis of EMPA-KIDNEY showed that empagliflozin slowed the rate of progression of chronic kidney disease among all types of participant in the trial, suggesting that albuminuria alone should not be used to determine whether to treat with an SGLT2 inhibitor [101]. The results of renal primary endpoint trials involving SGLT2i are summarized in Table 2.

Based on the nephroprotection data by RCTs, the American Diabetes Association’s 2024 Standards of Care recommended, for the management of patients with type 2 diabetes and CKD, that an SGLT2i with documented evidence of reduced CKD progression is preferred [102]. In particular, it is suggested to undertake glycosuric therapy when the GFR ≥ 20 mL/min/1.73 m^2^ and, once started, to continue it until the start of dialysis or transplantation. In addition, the approval of dapagliflozin and empagliflozin for the treatment of chronic kidney disease regardless of the presence of diabetic disease, has added a new option for the management of CKD. Based on the results of randomized clinical trials, SGLT2 inhibitors are now recommended as first-line therapy along with RAAS blockers in the majority of patients with CKD [82]. In addition to randomized clinical trials, secondary analyses and real-life studies have also confirmed the nephroprotective results of SGLT2i. A recent meta-analysis that included 13 studies with more than 90,000 participants with a large mean baseline eGFR range of 37–85 mL/min/1.73 m^2^ and of which 82.7% had T2DM showed that SGLT2i reduced the risk of kidney disease progression by 37%, with similar effects in patients with DKD or other causes of CKD, also demonstrating a statistically significant reduction in the risk of cardiovascular death [103]. Among the real-life experiences, one of the key studies was CVD-REAL3, which included more than 65,000 patients, spread across the entire spectrum of eGFR and albuminuria, and showed a 51% reduction in the relative risk of a decrease of more than 50% in eGFR or the development of ESKD in patients treated with SGLT2i [104].

The evidence described supports the pleiotropic effects of SGLT2i, beyond the mere hypoglycemic effect, demonstrating the validity of drug repurposing of this pharmacological class in nephroprotection.

## 7. Conclusions

SGLT-2 inhibitors, as multifunctional drugs, have demonstrated solid renal and cardiovascular protective effects in the treatment of non-diabetic CKD, significantly expanding their clinical application and value [105]. Through multiple mechanisms, SGLT-2 inhibitors effectively reduce glomerular pressure, decrease proteinuria, and significantly delay the progression of kidney function deterioration [106]. These effects have been thoroughly validated in key clinical trials such as DAPA-CKD [8], CREDENCE [61], and EMPA-KIDNEY [100], demonstrating their broad applicability and significant efficacy across different patient populations.

First, SGLT2is regenerate tubulo-glomerular feedback, significantly lowering glomerular pressure and filtration rate, thereby alleviating the mechanical damage caused by hyperfiltration [107]. This effect is present in both diabetic and non-diabetic CKD patients, demonstrating its broad applicability among different pathological conditions. Additionally, SGLT2is reduce proteinuria, alleviating the pressure on the glomerular membrane and inhibiting related inflammatory and fibrotic responses, with a consequent protection of renal structure and function [53]. Secondly, the anti-inflammatory and antifibrotic mechanisms of SGLT2 inhibitors offer a fascinating perspective for their use in CKD treatment. By regulating key signaling pathways such as AMPK [108], HIF-1α, TGF-β, and NF-κB, SGLT-2 inhibitors effectively reduce inflammatory and fibrotic responses, thereby delaying kidney function deterioration [109]. These effects contribute to mitochondrial morphologic maintenance, DNA integrity, and participation in metabolic processes [18]. The synergistic action of these multiple mechanisms enhances the drug protective effects in CKD and provides potential applications for the treatment of other related diseases. Moreover, SGLT-2 inhibitors reduce blood pressure, uric acid levels, and body weight, improving patients’ metabolic state and significantly lowering cardiovascular disease risk [110]. This global effect improves patients’ quality of life and reduces the healthcare burden of cardiovascular diseases. Indeed, SGLT2is represent one of the four pillars of heart failure (HF) pharmacological therapy and are recommended by current guidelines in all patients with HF, regardless of the left ventricular ejection fraction [111]. Moreover, even if the anti-arrhythmic effect of SGLT2i has not been clearly demonstrated yet, a recent real-world study involving HF patients with reduced ejection fraction and cardiac implantable electronic devices showed that the use of SGLT2is was associated with significant reduction in terms of atrial and ventricular arrhythmias [112].

Although SGLT-2 inhibitors have demonstrated significant efficacy and broad potential in diabetic and non-diabetic CKD patients, their use requires careful management of potential side effects, such as urinary tract infections, hypotension, and diabetic ketoacidosis [113]. These risks can be effectively controlled and managed through appropriate patient selection, regular monitoring, and individualized treatment plans. High-risk populations, such as elderly patients, children, and those with a history of infections or metabolic disorders, require special attention and preventive measures to ensure the safe use of the drug.

By combining SGLT-2 inhibitors with RAAS blockers and antifibrotic drugs, significant synergistic effects are achieved, enhancing renal protection and delaying the progression of kidney disease [114]. This multidrug combination therapy excels in controlling blood pressure and proteinuria and shows synergistic advantages in inhibiting inflammation and fibrosis, offering a more comprehensive and effective treatment plan for CKD patients.

In the future, further exploration of the mechanisms of SGLT-2 inhibitors and long-term safety and efficacy evaluations in different populations will be key to promoting their widespread use. The well-established benefits of SGLT2is in nontransplant populations have sparked interest in their potential applications for kidney transplant recipients; however, the actual evidence on the safety and efficacy of SGLT2i therapy in this kind of population is very limited [115]. Additionally, it is plausible and reasonable to speculate that SGLT2is may have a positive effect on kidney function during an episode of AKI through their anti-inflammatory and anti-oxidative effects [116]. Likely for the same reasons, SGLT2is have demonstrated efficiency in different scenarios and kidney pathologies that were unexpected, such as IgA nephropathy and focal and segmental glomerulosclerosis [116]. Finally, from a holistic view, the expression of SGLT2 not only in the kidney but also in several different tissues (e.g., brain, lung, and testis) might explain the emerging evidence of SGLT2is in cognitive, respiratory, and sexual dysfunction [117].

Certainly, SGLT2is are at the moment a pivotal drug choice in the management of CKD and related chronic conditions. In fact, these compounds significantly delay kidney function deterioration through multiple mechanisms and improve overall metabolic and cardiovascular health, enhancing patients’ quality of life and prognosis.

## Figures and Tables

**Figure 1 biomedicines-13-02123-f001:**
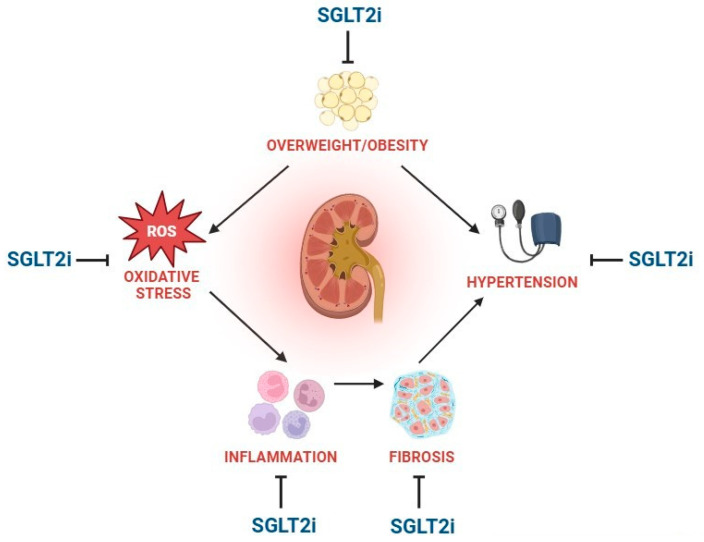
The figure shows the nephroprotective mechanisms offered by SGLT2is. The middle of the figure represents the pathogenetic interplay involved in chronic kidney disease (overweight/obesity; oxidative stress; inflammation; fibrosis; and hypertension). The outer regions represent the inhibition signals conveyed by SGLT2is on all these specific processes of kidney damage.

**Table 2 biomedicines-13-02123-t002:** The table shows the results of renal primary endpoint trials involving SGLT2i described in the text. ESRD, end-stage renal disease; eGFR, estimated glomerular filtration rate.

RCT, Year	Intervention	Kidney Outcome	Findings	Reference
CREDENCE, 2019	Canagliflozin vs. Placebo	ESRD, doubling of the serum creatinine level, or death from renal or cardiovascular causes	Risk reduction of 30%	[61]
DAPA-CKD, 2020	Dapagliflozin vs. Placebo	Sustained ≥ 50% reduction in eGFR, ESRD, or death from renal or cardiovascular cause	Risk reduction of 39%	[8]
EMPA-KIDNEY, 2023	Empagliflozin vs. Placebo	ESRD, a sustained reduction in eGFR to <10 mL/min/1.73 m^2^, renal death, or a sustained decline ≥ 40% in eGFR	Risk reduction of 28%	[100]

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
