# Peer review of "Nephroprotective Mechanisms of SGLT2i: Beyond the Glucose-Lowering Effect"

_biomedicines, 2025, doi:10.3390/biomedicines13092123_

Round 1
Reviewer 1 Report
Comments and Suggestions for Authors
Congratulations to the authors for their interesting manuscript; I think that there is room for improvement in some of its areas:
Although the manuscript addresses metabolic effects, the discussion could be strengthened by elaborating on the role of mitochondrial biogenesis and the modulation of renal hypoxia-inducible factors (such as HIF-1α, HIF-2α). In addition, the interaction between SGLT2 inhibitors and RAAS blockade, particularly regarding their differential effects on afferent and efferent arteriolar tone, should be clarified.
The authors should also highlight emerging evidence that the nephroprotective benefits of SGLT2 inhibitors extend to normoalbuminuric CKD, as demonstrated in subgroup analyses from trials such as EMPA-KIDNEY. Moreover in order to expand their discussion authors are encouraged to include other pleiotropic effects of SGLT2i (doi: 10.1002/ehf2.15223)
The scope of the review could be broadened by integrating recent data on SGLT2 inhibitor use in kidney transplant recipients, patients with glomerular diseases such as IgA nephropathy and focal segmental glomerulosclerosis, and in the prevention of acute kidney injury.
From a presentation standpoint, authors should consider including a concise summary table of major randomized controlled trials detailing study populations, inclusion criteria, primary endpoints, and relative risk reductions would provide a valuable quick-reference tool for clinicians.
Author Response
Dear Reviewer 1,
We appreciate your positive feedback and your thoughtful comments. Following, we report a point-to-point response to all your suggestions. All changes in text are marked in red.
- Although the manuscript addresses metabolic effects, the discussion could be strengthened by elaborating on the role of mitochondrial biogenesis and the modulation of renal hypoxia-inducible factors (such as HIF-1α, HIF-2α)
In addition to the already described role of HIF-1α, we delved into the action of SGLT2i on HIF-2α (Line 192-195).
- In addition, the interaction between SGLT2 inhibitors and RAAS blockade, particularly regarding their differential effects on afferent and efferent arteriolar tone, should be clarified.
We better explained the topic of the interaction between SGLT2 inhibitors and RAAS blockade (Line 285-288).
- The authors should also highlight emerging evidence that the nephroprotective benefits of SGLT2 inhibitors extend to normoalbuminuric CKD, as demonstrated in subgroup analyses from trials such as EMPA-KIDNEY.
We cited the requested secondary analysis of EMPA-KIDNEY. (Line 404-408)
- Moreover in order to expand their discussion authors are encouraged to include other pleiotropic effects of SGLT2i (doi: 10.1002/ehf2.15223)
In the conclusion section, we added a paragraph about the antiarrhythmic effects of SGLT2i and we cited the suggested reference (Line 466-472)
- The scope of the review could be broadened by integrating recent data on SGLT2 inhibitor use in kidney transplant recipients, patients with glomerular diseases such as IgA nephropathy and focal segmental glomerulosclerosis, and in the prevention of acute kidney injury.
In the conclusion section, we added a paragraph about the possible use of SGLT2i in these specific populations (Line 489-497)
- From a presentation standpoint, authors should consider including a concise summary table of major randomized controlled trials detailing study populations, inclusion criteria, primary endpoints, and relative risk reductions would provide a valuable quick-reference tool for clinicians.
We added a table about the major randomized controlled trials cited in the text (Table 2.)
Reviewer 2 Report
Comments and Suggestions for Authors
Clinically relevant and scientifically sound review of "Nephroprotective mechanisms of SGLT2 inhibitors beyond glucose".
A general comment is that words like "game-changer", "exceptional" or similar may not feel fully appropriate for a scientific review paper, please consider appropriate rewording.
Please find my specific comments below:
line 43: please note that SGLT2 inhibitors do NOT necessarily "induce normalization of blood glucose", please consider appropriate rewording
line 44: in the same sentence, please note that SGLT2 inhibitors do not lead to "progressive" weight loss, please consider appropriate rewording
line 46: SGLT2 inhibitors do NOT "reverse" pathophysiology, please consider appropriate rewording
line 81-83: please note that the glucagon effect may not be the only or even most relevant explanation here, especially in the context of glucagon AGONISTS that are now included in weight loss drugs, cf. retatrutide. Please consider adjustment of this statement
line 98-106 "effect on dyslipidemia", please contextualize appropriately and mention that SGLT2 inhibitors may also lead to minor increases of cholesterol, which has been described extensively.
line 107-132: authors may want to consider a more balanced perspective regarding the ketone and metabolic reprogramming paragraph. There aren't only positive aspects of ketogenesis. Risks may include cellular senescence (Wei et al, Sci Adv. 2024 May 17;10(20)) and the risk of ketoacidosis is mentioned in the SGLT2i labels, see specific comment below
line 113: please consider refering to the risk of ketoacidosis later in your review, especially in the context of insulinopenia
line 172-222: authors may want to consider taking the recent overview regarding blood pressure and SGLT2 into consideration: Ahwin, P., Martinez, D. The relationship between SGLT2 and systemic blood pressure regulation. Hypertens Res 47, 2094–2103 (2024)
line 234-235: "In patients with T2DM, SGLT2i therapy induces an acute dose- 233
dependent reduction in eGFR of approximately 5 ml/min/1.73 m2 in the initial weeks, then returning to baseline and stabilizing over time [45,52]" This statement is not correct, both regarding the magnitude and also regarding the return to baseline. There is a hemodynamic effect of eGFR reduction in the early period after SGLT2i initiation, BUT this only returns to "pre-SGLT2i" when you go off treatment. This is also why the FDA requires and off-treatment measurement of eGFRs after stopping the drug. Notably, the reference and citation is actually INCORRECT. Please adjust accordingly
line 259-265: The authors may want to consider a more elaborate discussion on AKI. Especially since AKI reporting (as AE) may not be according to KDIGO standards (Lameire et al. Kidney Int. 2021 Sep;100(3):516-526). Alternatively, to avoid misunderstandings, the AKI paragraph could also be deleted since it does not add significant additional value
line 307 ff: authors may want to consider going beyond DKD or CKD and mention the "general" nephroprotective mechanism for other additional indications, e.g., IgA nephropathy BUT highlight as a limitation that this effect is usually not addressing the causative mechanism
lines 383-386: authors may want to adjust the sentence "...significantly delay the progression of kidney function deterioration through anti-inflammatory and antifibrotic effects. [81]." so that it becomes clear that the benefit is not only due to the anti-inflammation and anti-fibrotic effects but via the nicely described totality of MoA/evidence.
Author Response
Dear Reviewer 2,
We appreciate your positive feedback and your thoughtful comments. Following, we report a point-to-point response to all your suggestions. All changes in text are marked in red.
- A general comment is that words like "game-changer", "exceptional" or similar may not feel fully appropriate for a scientific review paper, please consider appropriate rewording.
As you suggested, we chose appropriate rewording in the text.
- line 43: please note that SGLT2 inhibitors do NOT necessarily "induce normalization of blood glucose", please consider appropriate rewording
We rephrased this sentence (Line 41).
- line 44: in the same sentence, please note that SGLT2 inhibitors do not lead to "progressive" weight loss, please consider appropriate rewording
We rephrased this sentence (Line 42).
- line 46: SGLT2 inhibitors do NOT "reverse" pathophysiology, please consider appropriate rewording
We rephrased this sentence (Line 45).
- line 81-83: please note that the glucagon effect may not be the only or even most relevant explanation here, especially in the context of glucagon AGONISTS that are now included in weight loss drugs, cf. retatrutide. Please consider adjustment of this statement
Glucagon as all the other counterregulatory hormones is known to raise blood sugar and induce weight gain. Specifically, retatrutide is a trial agonist where the weight loss effect is mediated by the simultaneous action of GlP, GLP-1, and glucagon receptor agonist. Anyway we rephrased this sentence from doubtful perspective (Line 98).
- line 98-106 "effect on dyslipidemia", please contextualize appropriately and mention that SGLT2 inhibitors may also lead to minor increases of cholesterol, which has been described extensively.
We rephrased this sentence (Line 114).
- line 107-132: authors may want to consider a more balanced perspective regarding the ketone and metabolic reprogramming paragraph. There aren't only positive aspects of ketogenesis. Risks may include cellular senescence (Wei et al, Sci Adv. 2024 May 17;10(20)) and the risk of ketoacidosis is mentioned in the SGLT2i labels, see specific comment below
We addressed the risk of ketoacidosis related to SGLT2i both in the “safety profile” and in the “ketogenesis” paragraphs. Moreover we added the suggested reference (Line 150-154).
- line 113: please consider refering to the risk of ketoacidosis later in your review, especially in the context of insulinopenia
In order to better explain the topic of ketoacidosis, we preferred to explore it in both paragraphs (sub-paragraph 2.3 and section 5).
- line 172-222: authors may want to consider taking the recent overview regarding blood pressure and SGLT2 into consideration: Ahwin, P., Martinez, D. The relationship between SGLT2 and systemic blood pressure regulation. Hypertens Res 47, 2094–2103 (2024)
We cited the suggested reference and we added a new sub-paragraph (Line 300-315)
- line 234-235: "In patients with T2DM, SGLT2i therapy induces an acute dose- 233 dependent reduction in eGFR of approximately 5 ml/min/1.73 m2 in the initial weeks, then returning to baseline and stabilizing over time [45,52]" This statement is not correct, both regarding the magnitude and also regarding the return to baseline. There is a hemodynamic effect of eGFR reduction in the early period after SGLT2i initiation, BUT this only returns to "pre-SGLT2i" when you go off treatment. This is also why the FDA requires and off-treatment measurement of eGFRs after stopping the drug. Notably, the reference and citation is actually INCORRECT. Please adjust accordingly
We are grateful for your suggestion and we added the correct reference to the paragraph. (Line 263-265) We decided to report the sentence as written in the cited article “and slowly returning towards pretreatment values over the next 3–9 months”.
- line 259-265: The authors may want to consider a more elaborate discussion on AKI. Especially since AKI reporting (as AE) may not be according to KDIGO standards (Lameire et al. Kidney Int. 2021 Sep;100(3):516-526). Alternatively, to avoid misunderstandings, the AKI paragraph could also be deleted since it does not add significant additional value reported in AE the minimal frequency of aki
We apologize but we preferred to maintain the discussion on the risk of AKI, as we think it is essential to know that AKI is a not so frequent but severe adverse effect related to SGLT2i use.
- line 307 ff: authors may want to consider going beyond DKD or CKD and mention the "general" nephroprotective mechanism for other additional indications, e.g., IgA nephropathy BUT highlight as a limitation that this effect is usually not addressing the causative mechanism conclusion section
We added this topic in the conclusion section (Line 489-497)
- lines 383-386: authors may want to adjust the sentence "...significantly delay the progression of kidney function deterioration through anti-inflammatory and antifibrotic effects. [81]." so that it becomes clear that the benefit is not only due to the anti-inflammation and anti-fibrotic effects but via the nicely described totality of MoA/evidence.
We rephrased this sentence (Line 443).
Reviewer 3 Report
Comments and Suggestions for Authors
- The abstract currently lacks a clearly articulated objective, which makes it difficult for the reader to immediately grasp the central focus and intended scope of the review. Incorporating a concise and well-defined aim would significantly enhance clarity and guide the reader through the thematic direction of the article.
- In the introduction section, while the therapeutic class of SGLT2 inhibitors is discussed in detail, specific examples of drugs are not mentioned. Including these well-known agents would enhance the paragraph's clarity and make it more informative for readers who may not be familiar with this drug class.
- The author should consistently refer to SGLT2i as inhibitors throughout the manuscript. In several instances, terms like “molecules” or other non-specific descriptors are used, which may create confusion for readers. Maintaining uniform terminology will enhance clarity and ensure scientific accuracy, especially considering the focus of the review is specifically on SGLT2 inhibitors.
- The introduction would benefit from the inclusion of epidemiological data regarding the global burden of chronic kidney disease (CKD) and heart failure (HF), especially in diabetic populations. Citing WHO or other relevant health statistics would provide stronger justification for the clinical significance of SGLT2 inhibitors.
- The introduction does not clearly define the existing knowledge gaps or state the specific aim of the review. To enhance clarity and focus, the author should briefly highlight what is lacking in current literature and include a clear objective.
- Figure 1 appears to be drawn in a very simplistic manner, seemingly just to fulfill formal requirements. It lacks any mechanistic detail and merely shows “inhibition” of various factors without specifying how SGLT2 inhibitors act. This does not meet the standard of a scientific figure. Additionally, the visible "BioRender" label below the image suggests that the figure was prepared casually without customization, which reflects carelessness in presentation.
- The ligands presented in Figure 1 require better explanation. While the figure is briefly mentioned in the introduction, no critical discussion about its components or their relevance has been provided in the text.
- In line 68 The phrase “thanks to” is informal and more suitable for conversational writing. Since this is a scientific or academic context, consider using a more formal and precise alternative.
- In lines 85–90 the paragraph references studies by Xu et al. and another involving ApoE-/- mice, both of which are animal studies, yet the transition from general mechanisms to preclinical (animal) evidence is not clearly stated. This can lead to confusion, especially if readers assume the effects described (e.g., reduced inflammation, increased adiponectin) are from human trials.
- Line 91-93 It is suggested that SGLT2i reduces sympathetic nervous system activity, but the clinical relevance of this moderate weight loss to SNS activity is not well explained consider elaborating.
- Lines 94–96 Potential tonal contradiction: The use of “modest” and “significantly contributing” in the same sentence may appear contradictory unless clarified. Consider explaining that the weight loss is modest in amount but significant in metabolic or clinical impact.
- In line 98-100 metabolic shift from glucose metabolism to free fatty acid oxidation due to SGLT2i treatment. However, it lacks an explanation of the underlying mechanism by which SGLT2 inhibitors induce this shift. Need to Add a brief clarification.
- In line 105, it is mentioned 'with beneficial qualitative changes on LDL and HDL particles' what kind of qualitative changes?
- Lines 106–108, “SGLT2 inhibitors stimulate FFA oxidation and ketone production”, but it does not explain how this occurs at the molecular level. There is no mention of specific signaling pathways or targets.
- Lines 109–110, The text states that ketone levels can rise to the millimolar range, but it lacks any specific numerical values or supporting clinical study references. This makes it difficult to assess the actual extent of ketone elevation.
- GLUT9b is mentioned as a transporter involved in reduced urate uptake, but the underlying mechanism how SGLT2 inhibition affects GLUT9b function or expression is not explained.
- Lines 137–140, A meta-analysis is mentioned with findings on uric acid reduction, but no specific study name, author, or publication source is provided only a numeric reference is cited.
- The paragraph “Antioxidant, anti-inflammatory and antifibrotic effects” attempts to explain multiple mechanisms like oxidative stress, nitric oxide signaling, RAAS inhibition, AGEs suppression, HIF-1α, and NF-kB pathways all within a single block of text. Need to Break this content into 2–3 shorter paragraphs, each focusing on a specific mechanism. This will improve clarity and readability for the reader.
- Consider including a schematic and mechanistic figure that visually summarizes these pathways. This will enhance reader understanding and engagement.
- Provide appropriate references to support the statements made between lines 160-170.
- In several places throughout the manuscript, references are missing. For example, the paragraph from lines 345 to 365 has been written without any citations. The author should ensure that such gaps are addressed and appropriate references are provided.
- Throughout the manuscript, the author has discussed various pathways and mechanisms in detail. To enhance the clarity and scientific presentation of the manuscript, it is advisable to include at least two figures and one or two tables. These additions will assist in effectively conveying complex information and improving the overall understanding of the content.
Author Response
Dear Reviewer 3,
We appreciate your positive feedback and your thoughtful comments. Following, we report a point-to-point response to all your suggestions. All changes in text are marked in red.
- The abstract currently lacks a clearly articulated objective, which makes it difficult for the reader to immediately grasp the central focus and intended scope of the review. Incorporating a concise and well-defined aim would significantly enhance clarity and guide the reader through the thematic direction of the article.
We appreciate your suggestion and we clarified the aim of our review in the abstract and at the end of the introduction section.
- In the introduction section, while the therapeutic class of SGLT2 inhibitors is discussed in detail, specific examples of drugs are not mentioned. Including these well-known agents would enhance the paragraph's clarity and make it more informative for readers who may not be familiar with this drug class.
We added specific examples of SGLT2i to help the reader become more familiar with this drug class (Line 51-54)
- The author should consistently refer to SGLT2i as inhibitors throughout the manuscript. In several instances, terms like “molecules” or other non-specific descriptors are used, which may create confusion for readers. Maintaining uniform terminology will enhance clarity and ensure scientific accuracy, especially considering the focus of the review is specifically on SGLT2 inhibitors.
As you suggested, we substitute all the non-specific words to avoid possible misunderstandings among readers.
- The introduction would benefit from the inclusion of epidemiological data regarding the global burden of chronic kidney disease (CKD) and heart failure (HF), especially in diabetic populations. Citing WHO or other relevant health statistics would provide stronger justification for the clinical significance of SGLT2 inhibitors.
We added the suggested data (Line 45-48).
- The introduction does not clearly define the existing knowledge gaps or state the specific aim of the review. To enhance clarity and focus, the author should briefly highlight what is lacking in current literature and include a clear objective.
We better highlighted the aim of the review at the end of the introduction section (Line 77-79). Moreover we preferred to discuss the existing knowledge gaps in the conclusion section.
- Figure 1 appears to be drawn in a very simplistic manner, seemingly just to fulfill formal requirements. It lacks any mechanistic detail and merely shows “inhibition” of various factors without specifying how SGLT2 inhibitors act. This does not meet the standard of a scientific figure. Additionally, the visible "BioRender" label below the image suggests that the figure was prepared casually without customization, which reflects carelessness in presentation.
We customized the figure by removing the “Biorender” label below the image. We understand your criticisms, but we intentionally draw the figure as simple as possible in order to be immediately understandable even for the non expert reader. Following the figure, we added a table (Table 1.) to explain in detail all the nephroprotective mechanisms.
- The ligands presented in Figure 1 require better explanation. While the figure is briefly mentioned in the introduction, no critical discussion about its components or their relevance has been provided in the text.
We improved the description of Figure 1 in the caption. (Line 67-70)
- In line 68 The phrase “thanks to” is informal and more suitable for conversational writing. Since this is a scientific or academic context, consider using a more formal and precise alternative.
We rephrased this sentence. (Line 83)
- In lines 85–90 the paragraph references studies by Xu et al. and another involving ApoE-/- mice, both of which are animal studies, yet the transition from general mechanisms to preclinical (animal) evidence is not clearly stated. This can lead to confusion, especially if readers assume the effects described (e.g., reduced inflammation, increased adiponectin) are from human trials.
We rephrased this sentence (Line 101).
- Line 91-93 It is suggested that SGLT2i reduces sympathetic nervous system activity, but the clinical relevance of this moderate weight loss to SNS activity is not well explained consider elaborating.
Broadly speaking the net weight loss effect of SGLT2i is modest. Moreover, clear data on the suggested topic are still lacking, so we preferred not to explore it in detail. (Line 108)
- Lines 94–96 Potential tonal contradiction: The use of “modest” and “significantly contributing” in the same sentence may appear contradictory unless clarified. Consider explaining that the weight loss is modest in amount but significant in metabolic or clinical impact.
We rephrased this sentence (Line 110-111)
- In line 98-100 metabolic shift from glucose metabolism to free fatty acid oxidation due to SGLT2i treatment. However, it lacks an explanation of the underlying mechanism by which SGLT2 inhibitors induce this shift. Need to Add a brief clarification.
We preferred not to explain in detail this shift in the “ dyslipidemia” paragraph as we think it could be misleading for the reader. We explored this topic in the subsequent section about ketogenesis.
- In line 105, it is mentioned 'with beneficial qualitative changes on LDL and HDL particles' what kind of qualitative changes?
We clarified this sentence (Line 122-123).
- Lines 106–108, “SGLT2 inhibitors stimulate FFA oxidation and ketone production”, but it does not explain how this occurs at the molecular level. There is no mention of specific signaling pathways or targets.
We apologize but we think the description of the molecular pathway of ketogenesis is off-topic to the goal of our review.
- Lines 109–110, The text states that ketone levels can rise to the millimolar range, but it lacks any specific numerical values or supporting clinical study references. This makes it difficult to assess the actual extent of ketone elevation.
In the text, we added both the normal range of ketones levels and the levels reached under SGLT2i treatment (Line 134-135)
- GLUT9b is mentioned as a transporter involved in reduced urate uptake, but the underlying mechanism how SGLT2 inhibition affects GLUT9b function or expression is not explained.
We better explained this sentence (Line 159-161)
- Lines 137–140, A meta-analysis is mentioned with findings on uric acid reduction, but no specific study name, author, or publication source is provided only a numeric reference is cited.
We better specified the reference (Line 162).
- The paragraph “Antioxidant, anti-inflammatory and antifibrotic effects” attempts to explain multiple mechanisms like oxidative stress, nitric oxide signaling, RAAS inhibition, AGEs suppression, HIF-1α, and NF-kB pathways all within a single block of text. Need to Break this content into 2–3 shorter paragraphs, each focusing on a specific mechanism. This will improve clarity and readability for the reader.
We apologize but we preferred to maintain a single section about “Antioxidant, anti-inflammatory and antifibrotic effects” due to the strenghtful relationship between inflammation and fibrosis.
- Consider including a schematic and mechanistic figure that visually summarizes these pathways. This will enhance reader understanding and engagement.
We apologize but we preferred to add a table about all the nephroprotective mechanisms of SGLT2i (Table 1.)
- Provide appropriate references to support the statements made between lines 160-170.
We provided appropriate references as you suggested
- In several places throughout the manuscript, references are missing. For example, the paragraph from lines 345 to 365 has been written without any citations. The author should ensure that such gaps are addressed and appropriate references are provided.
We tried to address such gaps with the appropriate references.
- Throughout the manuscript, the author has discussed various pathways and mechanisms in detail. To enhance the clarity and scientific presentation of the manuscript, it is advisable to include at least two figures and one or two tables. These additions will assist in effectively conveying complex information and improving the overall understanding of the content.
Beyond Figure 1., we added Table 1 and Table 2 to simplify the reading of the manuscript.
Reviewer 4 Report
Comments and Suggestions for Authors
Thank you for the opportunity to review this paper, which examines the nephroprotective mechanism of SGLT2i. This paper addresses an interesting topic commonly encountered in internal medicine, and specifically in the nephrological practice of CKD patients. However, there are some points that need to be clarified.
1) The SGLT2i block the transporter protein called SGLT2 (sodium–glucose co-transporter 2), which is responsible for reabsorbing glucose from the tubular lumen.
Before examining the pleiotropic effects of these drugs, I suggest adding a chapter that shows the distribution of transporter SGLT2 in the kidney and extrarenal sites. A Table could also help the reader to understand which pleiotropic effect of this class of drugs is related to renal sites, and which is likely related to extrarenal interaction.
In effects, if it’s true that the proximal tubule epithelial cells (S1 segment of the proximal convoluted tubule) is the main site of action and the main reason SGLT2i- induced lower blood glucose, osmotic diuresis, increased urine sodium excretion, reduced volemia, antiinflammatory effect, tubular-glomerular feedback reset), low-level SGLT2 expression in other tissues such as brain, liver, thyroid, muscle, and heart has been reported.
Extrarenal expression of the transporter is minimal, and its role in pleiotropic effects is so far unclear.
2) This is a very good narrative review. However, as a concise summary for the reader, I suggest adding a Table summarizing the demonstrated activity (with its reference) for every discussed pleiotropic effect (metabolic effects, antioxidant, anti-inflammatory, and antifibrotic effects, hemodynamic effects).
Author Response
Dear Reviewer 4,
We appreciate your positive feedback and your thoughtful comments. Following, we report a point-to-point response to all your suggestions. All changes in text are marked in red.
- The SGLT2i block the transporter protein called SGLT2 (sodium–glucose co-transporter 2), which is responsible for reabsorbing glucose from the tubular lumen. Before examining the pleiotropic effects of these drugs, I suggest adding a chapter that shows the distribution of transporter SGLT2 in the kidney and extrarenal sites. A Table could also help the reader to understand which pleiotropic effect of this class of drugs is related to renal sites, and which is likely related to extrarenal interaction. In effects, if it’s true that the proximal tubule epithelial cells (S1 segment of the proximal convoluted tubule) is the main site of action and the main reason SGLT2i- induced lower blood glucose, osmotic diuresis, increased urine sodium excretion, reduced volemia, antiinflammatory effect, tubular-glomerular feedback reset), low-level SGLT2 expression in other tissues such as brain, liver, thyroid, muscle, and heart has been reported. Extrarenal expression of the transporter is minimal, and its role in pleiotropic effects is so far unclear.
Since the effects mediated by SGLT2i on extra-renal transporters remain unclear, we only added a sentence in the conclusion section (Line 497-500). As these pleiotropic effects play a marginal role as compared to current knowledge about SGLT2i, we apologize but we did not find suitable to add another table about them.
- This is a very good narrative review. However, as a concise summary for the reader, I suggest adding a Table summarizing the demonstrated activity (with its reference) for every discussed pleiotropic effect (metabolic effects, antioxidant, anti-inflammatory, and antifibrotic effects, hemodynamic effects).
We appreciate your suggestion and we added a table summarizing all the nephroprotective mechanisms of SGLT2i (Table 1.)
Round 2
Reviewer 1 Report
Comments and Suggestions for Authors
Congratulations to the authors for having implemented all of my comments in their manuscript.
Reviewer 3 Report
Comments and Suggestions for Authors
Authors addressed all my concerns/comments. Therefore, I would recommend this manuscript to accept for publications in the journal.
Reviewer 4 Report
Comments and Suggestions for Authors
I have no further comments